# Chemical Modifications to Enhance the Drug Properties of a VIP Receptor Antagonist (ANT) Peptide

**DOI:** 10.3390/ijms25084391

**Published:** 2024-04-16

**Authors:** Christina Lester, Jian-Ming Li, Tenzin Passang, Yuou Wang, Edmund K. Waller, Simon B. Blakey

**Affiliations:** 1Department of Chemistry, Emory University, Atlanta, GA 30322, USA; cmleste@emory.edu; 2Department of Hematology and Oncology, Emory University School of Medicine, Atlanta, GA 30322, USA; jli9@emory.edu (J.-M.L.); yuou.wang@emory.edu (Y.W.); 3Cancer Biology Graduate Program, Emory University, Atlanta, GA 30322, USA; tenzin.passang.fnu@emory.edu; 4Winship Cancer Institute, Emory University, Atlanta, GA 30322, USA

**Keywords:** VIP, PEGylation, peptide staple, acute myeloid leukemia

## Abstract

Antagonist peptides (ANTs) of vasoactive intestinal polypeptide receptors (VIP-Rs) are shown to enhance T cell activation and proliferation in vitro, as well as improving T cell-dependent anti-tumor response in acute myeloid leukemia (AML) murine models. However, peptide therapeutics often suffer from poor metabolic stability and exhibit a short half-life/fast elimination in vivo. In this study, we describe efforts to enhance the drug properties of ANTs via chemical modifications. The lead antagonist (ANT308) is derivatized with the following modifications: N-terminus acetylation, peptide stapling, and PEGylation. Acetylated ANT308 exhibits diminished T cell activation in vitro, indicating that N-terminus conservation is critical for antagonist activity. The replacement of residues 13 and 17 with cysteine to accommodate a chemical staple results in diminished survival using the modified peptide to treat mice with AML. However, the incorporation of the constraint increases survival and reduces tumor burden relative to its unstapled counterpart. Notably, PEGylation has a significant positive effect, with fewer doses of PEGylated ANT308 needed to achieve comparable overall survival and tumor burden in leukemic mice dosed with the parenteral ANT308 peptide, suggesting that polyethylene glycol (PEG) incorporation enhances longevity, and thus the antagonist activity of ANT308.

## 1. Introduction

The vasoactive intestinal polypeptide (VIP) is a 28-residue neuropeptide with potent anti-inflammatory activity and immunosuppressive effects on T cells [1,2]. The VIP receptors VIP-R_1_ and VIP-R_2_ are widely expressed throughout the body, with differential expression on the surfaces of immune cells [3,4]. Furthermore, the overexpression of VIP and its receptors was previously reported in breast, prostate, and lung cancers, wherein VIP promotes growth and metastasis in tumors [5,6].

We have recently developed a series of VIP-R antagonists dubbed ANT002-308. These peptides possess a modified, positively charged N-terminus. The ANT peptides enhance T cell proliferation in vitro and improve T cell-dependent anti-tumor response in murine models of acute myeloid leukemia (AML) [2]. Furthermore, treatment with ANT peptides concomitant with anti-PD-1-combination therapy enhances intra-tumoral T cell proliferation and increases tumor-antigen-specific T cells within the tumor microenvironment (TME) of pancreatic ductal adenocarcinoma (PDAC) tumors (Figure 1) [7,8]. Despite the promising anti-tumor properties of these VIP-R antagonist derivatives, peptide therapeutics commonly suffer from poor metabolic stability and consequently exhibit a short half-life and fast elimination in vivo [9]. These limitations pose a significant barrier to the utility of therapeutic peptides as competent drug candidates.

Nevertheless, the emergence of site-selective peptide modifications has enabled the enhancement of therapeutic peptide drug properties. Such chemical modifications have facilitated the introduction of several peptide drugs in active clinical development and preclinical studies. We aimed to utilize chemical modifications to enhance the drug properties of the lead VIP-R antagonist, ANT308.

For the design of our ANT308 derivatives (Figure 2), we employed three modifications: N-terminus capping, covalent stapling, and C-terminus PEGylation. Because the termini of peptides are susceptible to cleavage by exopeptidases [10], we hypothesized that the acetylation of the N-terminus will impart stability towards proteolytic activity. Because native VIP adopts an α-helical structure from residues 10–23 [11], we hypothesized that the incorporation of a covalent staple within this region would enhance stability by masking proteolytically sensitive amide bonds. Additionally, we proposed that the staple would enhance binding affinity by pre-emptively conforming the peptide into its active conformation, thus reducing the entropic penalty of folding as the peptide approaches its transmembrane receptor. Furthermore, we hypothesized that the incorporation of a large, inert polyethylene glycol (PEG) linker onto the solvent-exposed C-terminus of ANT308 would extend the peptide’s half-life circulation and increase metabolic stability.

## 2. Results and Discussion

At the onset of our investigation, we first sought to modify the sequence of ANT308 to add a covalent staple within the α-helical region. A previous report by Gill et al. found that when employing an all-hydrocarbon staple, the most optimal placement to augment VIP was at residues L13 and M17 [12]. The authors hypothesized that this placement was beneficial for the following reason: the incorporation of an aliphatic staple resulted in a >20-fold increase in potency relative to a lactam staple poised in the same position, despite the lactam staple enhancing α-helicity considerably more than the aliphatic staple (29.4% vs. 9.6%, respectively). The authors proposed that the hydrocarbon staple enhanced potency through favorable hydrophobic interactions between the staple and the VIP-R_2_ receptor. The hydrocarbon staple at this placement caused a 3-fold increase in % parent peptide remaining post-incubation with trypsin.

Although the authors originally utilized the hydrocarbon stapling methodology developed by Verdine et al. [13], we were concerned that the cross-metathesis reaction used to furnish the aliphatic staple requires the use of a ruthenium catalyst, which can be difficult to remove by chromatography and is incompatible with downstream in vivo analyses. Additionally, the metathesis reaction requires the use of expensive non-canonical amino acids that are difficult to incorporate in solid-phase peptide synthesis (SPPS). Instead, we opted to use the bis-alkylation strategy developed by Meng et al. that utilizes canonical cysteine residues under mild and metal-free conditions [14]. Using this technology, we introduced a hydrophobic covalent staple at residues 13 and 17 using cysteine-selective bis-thiolalkylation (Figure 3).

After the incorporation of the *m*-xylene staple into ANT308, we examined the influence of the staple on the secondary structure via circular dichroism (CD) analysis. The CD spectra of the cysteine-substituted linear control (ANT308C13C17 unstp) closely resembles linear ANT308 (Figure 4). However, the stapled variant (ANT308C13C17 stp) exhibits a 2-fold increase in α-helicity relative to both linear peptides (Table 1). Furthermore, the incorporation of an *m*-xylene staple onto ANT308 exhibits similar helicity to the hydrocarbon staple reported for VIP at the same position (10.6% vs. 9.6%, respectively) [12]. These results are consistent with the literature, which shows that the *m*-xylene staple imparts slight-to-moderate helix induction and allows a degree of flexibility in aqueous media. While we did not consider additional solvents/buffers in this study, research indicates that *m*-xylene-stapled peptides exhibit enhanced α-helicity in solvents with additives that mimic membrane conditions or promote helix induction [15,16,17].

We next sought to incorporate PEG onto the C-terminus of ANT308. A previous report by DeRome et al. found that a 22 kDa PEG linker could be attached at the C-terminus of the VIP without disrupting the binding affinity to VIP-R_1_ [19]. These VIP-PEG conjugates were synthesized via the incorporation of cysteine onto the C-terminus of the VIP, which then reacts with maleimide-PEG. Because cysteine–maleimide chemistry is susceptible to thiol exchange in the presence of exogenous thiols [20], potentially leading to decomposition in biological media, we chose to employ strain-promoted copper-free click chemistry to generate our desired conjugates [21]. ANT308-PEG conjugates were synthesized via the incorporation of an azido-lysine at the C-terminus of ANT308 via SPPS, followed by a reaction with a 20 kDa dibenzocyclooctyne (DBCO)-m-PEG linker (Figure 5). Because PEG is neutral, excess PEG was removed by cation exchange chromatography (CEX) to furnish purified conjugates.

With the desired ANT308 derivatives in hand, we tested them for in vitro T cell activation. ANT308 results in a significant increase in T cell activation relative to the controls, whereas ANT308C13C17 unstp and ANT308C13C17 stp lead to a slight increase (Figure 6). Acetylated ANT308 loses the ability to augment T cell activation, indicating that the conservation of the N-terminus is critical for the antagonist activity of ANT308.

Next, we assessed ANT308-PEG for in vitro human T cell activation (Figure 7A–I); 10 μM of ANT308 and ANT308-PEG significantly enhanced CD69 expressions in CD4+ and CD8+ T cells. However, at a low concentration (1 μM), only ANT308-PEG significantly increased CD69+ populations (Figure 7A–C). A 10-fold increase in ANT308 concentration correlates to an increase in T cell proliferation, whereas ANT308-PEG exhibits similar proliferation regardless of dose. Ki67 is a T cell proliferation marker, and Granzyme is the most abundant serine protease stored in secretory granules of cytotoxic T cells (CTLs) and natural killer (NK) cells, functioning as one of the central factors in antitumor immunity. Here, at a single dose of 1 μM, we showed that ANT308-PEG more potently enhances Granzyme B and Ki67 expressions in CD4+ and CD8+ T cells compared to ANT308 (Figure 7D–I). Thus, PEGylation improved ANT308 peptide functions in vitro.

Acute myeloid leukemia (AML) blasts have been shown to create an immunosuppressive environment through the inhibition of T cell proliferation. The active suppression of T cell responses by AML blasts suggests that novel immunotherapeutics may play a role in the treatment of AML [1,22]. Thus, the stapled and PEGylated ANT308 derivatives were administered to AML and T cell leukemia-bearing mice to evaluate their immunotherapeutic potential. The survival rate and tumor size of mice treated with 10 doses of ANT308 or its derivatives over a 60-day period were measured (Figure 8A,B).

Mice treated with ANT308 show significantly higher survival rates and lower tumor burden relative to the phosphate buffered saline (PBS) control. In contrast, treatment with ANT308C13C17 unstp only slightly enhances overall survival and leads to a tumor burden comparable to the PBS control. This indicates that the replacement of L13/M17 with cysteine considerably diminishes the in vivo anti-cancer activity of ANT308. Despite this, ANT308C13C17 stp shows a moderate enhancement in overall survival and reduces tumor burden relative to ANT308C13C17 unstp, suggesting that the incorporation of a covalent staple at residues 13 and 17 enhances antagonist activity. Despite this improvement, by day 50, the treatment with a parenteral ANT308 peptide led to a slight enhancement of overall survival and reduced tumor burden relative to treatment with ANT308C13C17 stp.

Finally, we compared the survival rate of mice treated with varying doses of ANT308-PEG over a 60-day period to those treated with unmodified ANT308 (Figure 9A,B). Mice treated with ANT308-PEG and ANT308 show a significant enhancement of survival relative to the scrambled peptide control. Furthermore, four doses of ANT308-PEG given twice a week for two weeks prolonged overall survival comparable to fourteen doses of ANT308. This suggests that the incorporation of PEG onto ANT308 increases the longevity of the peptide, thereby having a comparable effect at a lower dosage.

## 3. Materials and Methods

### 3.1. Materials

Canonical N-α-Fmoc-*L*-amino acids were obtained from Oakwood Chemical, Estill, SC, USA. Fmoc-Lys(N_3_)-OH was purchased from Aapptec, Louisville, KY, USA, and Fmoc-Asp(OMpe)-OH was purchased from Combi-Blocks, San Diego, CA, USA. NovaSyn^®^ TGR resin was obtained from Novabiochem, Merck Biosciences, Darmstadt, Germany. All peptides were prepared by solid-phase automated synthesis (CEM Liberty Blue, Matthews, NC, USA) using standard Fmoc protocol. Asparagine (N), cysteine (C), and glutamine (Q) were incorporated with Trt-protected side chains. Aspartic acid (D) was incorporated with an OMpe-protected side chain. Arginine (R) was incorporated with a Pbf-protected side chain. Lysine (K) was incorporated with a Boc-protected side chain. Serine (S), threonine (T), and tyrosine (Y) were incorporated with *^t^*Bu-protected side chains. DBCO-mPEG, 20 kDa was purchased from Click Chemistry Tools, Scottsdale, AZ, USA. Synthetic reagents and solvents of the highest grade were purchased from MilliporeSigma, Darmstadt, Germany, and used without further purification. Gel electrophoresis materials were purchased from BioRad, Hercules, CA, USA.

### 3.2. Linear Peptide Synthesis

All peptides were prepared as carboxamides at the C-terminus on NovaSyn^®^ (Hyderabad, Telangana, India) TGR resin (0.24 mmol/g). Amino acid stock solutions (0.2 M) were used, along with 1.0 M Oxyma Pure and 1.0 M DIC. All stock solutions were prepared using HPLC-grade DMF. For Fmoc deprotection, 20% (*v*/*v*) piperidine solution in DMF was prepared. For N-terminus acetylation, 20% (*v*/*v*) acetic anhydride in DMF was used. Syntheses were carried out under a 0.1 mmol scale. All amino acids were singly coupled at 90 °C for 2 min, except for arginine, which was doubly coupled. The instrument was set to deliver 6 equivalents of amino acids, etc. N-terminus acetylation was performed using 4 iterative couplings at 60 °C.

### 3.3. Cleavage and Purification of Crude Peptides

Crude peptides without methionine or cysteine were cleaved under air using a cocktail of TFA. For peptides containing methionine or cysteine, cleavage was performed under nitrogen using a cocktail of 94% HPLC-grade TFA, 2.5% H_2_O, 2.5% EDT, and 1% TIPS for 2 h. For peptides containing azidolysine, cleavage was performed under nitrogen using a cocktail of 92.5:2.5:2.5:2.5 (*w*/*w*/*w*/*w*) TFA:H_2_O:TIPS:DTT for 2 h [23]. The crude peptide solution was triturated with 10 CV of cold Et_2_O. The white solid was centrifuged at 4000 rpm for 10 min to form a pellet. After removal of Et_2_O, the solid was resuspended twice with Et_2_O then dried under a stream of nitrogen. The crude material was reconstituted in 20% MeCN in H_2_O (0.1% TFA) and filtered using a 0.45 μm syringe filter (PTFE). The crude solution was purified on an Agilent 1260 preparatory RP-HPLC using a Pursuit C18 column (21.2 mm × 250 mm, 5 μm) at 20 mL/min. The peptides were purified using H_2_O (solvent A) and MeCN (solvent B) with a 0.1% TFA co-solvent under the following gradient (Table 2).

Peptide elution was monitored via UV absorption at 220 nm. The desired fractions were collected and lyophilized to yield a purified product as a white powder, stored at −20 °C.

### 3.4. Peptide Stapling and Purification

Cysteine-derived peptides were prepared as a 10 mg/mL stock solution in 1:7 MeCN: 20 mM NH_4_HCO_3_ buffer (pH 7.8). Reactions were performed using 1 mL aliquots and 5 equivalents of 1,3-bis(bromomethyl)benzene. The peptide solutions were stirred at room temperature under nitrogen. The starting material consumption was monitored by an Agilent 1260 LC-MS using a Zorbax C18 column. After reaction completion, the crude material was purified on an Agilent 1260 semi-preparatory RP-HPLC using a Pursuit C18 column (10 mm × 250 mm, 5 μm) at 10 mL/min. A crude stapled peptide was purified using a gradient of 25–75% MeCN (0.1% TFA) over 25 min. The desired fractions were collected and lyophilized to yield a purified stapled product as a white powder, stored at −20 °C.

### 3.5. Synthesis and Purification of Peptide-PEG Conjugates

Azido-lysine-derived peptides were prepared as a 1 mg/mL solution in a 1× PBS buffer (pH 7.4). Two equivalents of DBCO-mPEG, 20 kDa, were added to the peptide solution and stirred at room temperature for 24 h. Reaction completion was confirmed via 4–20% SDS-PAGE and staining with Coomassie Bio-safe™ (Bio-Rad, Hercules, CA, USA). Peptide-PEG conjugates were dialyzed (2 kDa MWCO) into 100 mM CAPS buffer (pH 10.6), and then purified on a Bio-Rad FPLC (NGC 10) via CEX using a HiTrap^®^ SP Sepharose FF column (Cytiva, Marlborough, MA, USA). After eluting free PEG, the desired PEG–peptide was eluted using 100 mM CAPS buffer (pH 10.6) with 250 mM NaCl. PEG–peptide elution was monitored via UV absorption at 220 nm. Fractions containing PEG–peptide were distinguished from free PEG via Coomassie staining. The desired fractions were collected and lyophilized to yield a product as a white powder, stored at −20 °C.

### 3.6. Peptide Analysis and Characterization

Purified peptides were characterized by high-resolution mass spectrometry using a Thermo LTQ-FTMS instrument (Thermo Fisher Scientific, Waltham, MA, USA) or an Agilent 6545XT LC-QTOF (Agilent Technologies Inc., Santa Clara, CA, USA). The mass-to-charge ratios were used to determine the experimental mass of the peptide, which was verified against the calculated mass. Analytical RP-HPLC spectra were obtained using an Agilent 1260 series with a Pursuit C18 column (4.6 mm × 250 mm, 5 μm). Purified peptides were prepared in 5% MeCN in H_2_O (0.1% TFA), and then run on a 5–95% MeCN gradient (0.1% TFA) over 10 min at 1 mL/min. Peptide purity was determined to be >95% via integration at 220 nm.

### 3.7. Circular Dichroism Spectroscopy

Circular dichroism spectra were obtained in an inert atmosphere using a Jasco J-1500 spectrometer (Jasco, Oklahoma City, OK, USA) measuring from 260 to 190 nm in 20 mM sodium phosphate buffer (pH 7.0) at 20 °C. A 0.1 mm path length quartz cuvette was used, and spectra were obtained with 0.5 nm data pitch, 100 nm/min scan speed, a time constant (D.I.T.) of 1 s, and a mean (*n* = 3) accumulation for each spectrum. Peptides were prepared as 5 mg/mL stock solutions in buffer. A blank of 20 mM sodium phosphate buffer was used to subtract the background from the peptide spectra. Raw data were processed to reveal mean residue ellipticity [θ] by normalizing for peptide concentration, path length, and the number of amide bonds using the following formula:$${\left[\theta \right]}_{\mathrm{MRW}}=\frac{(\mathrm{MRW}*\theta_{\mathrm{obs}})}{10*d*c}\;\deg\cdot {\mathrm{cm}}^{2}\cdot {\mathrm{dmol}}^{-1}$$

MRW or mean residue weight = $\frac{\mathrm{molecular}\;\mathrm{weight}\;(g/\mathrm{mol})}{(N-1)}$ where N is number of residues

θ_obs_ = observed ellipticity in degrees

d = pathlength in cm

c = concentration in g/mL.

### 3.8. α-Helical Content

Estimation of peptide secondary structure from circular dichroism spectra was performed using the web-based application BeStSel [18], a freely accessible tool for the estimation of peptide helicity. The α-helical content of the peptides qualitatively correlated with the CD spectra reported in this study.

### 3.9. In Vitro T Cell Studies

A leukapheresis product containing peripheral blood mononuclear cells (PBMC) from multiple healthy donors were obtained from Stem Cell Technologies. T cells were isolated from PBMC using human pan-T cell isolation kit, according to the manufacturer’s protocol (Miltenyi Biotec, Bergisch Gladbach, Germany, Catalog No. 130-096-535). Isolated T cells from 4–5 donors were pooled and seeded at a density of 1 × 10^6^/mL in 100 μL media in round-bottom wells in a 96-well plate, activated with a 1.5 μL/mL CD3/CD28 T cell activator (ImmunoCult, San Diego, CA, USA) in the presence of 30 IU interleukin 2 (IL2). Pooled T cells were activated in the presence or absence of peptides (scrambled, ANT308, or modified ANT308) and cultured for 48 h. Leukocyte Activation Cocktail with Golgi Plug (BD, Franklin Lakes, NJ, USA) was added 4 h prior to cell harvesting to assess Granzyme B expression in CD4+ and CD8+ T cells. Briefly, cells were stained with Fixable Aqua live/dead viability stain for 5 min at room temperature (RT). Surface antibodies (Table 3) were added to the cells at the desired concentration and left to stain for 30 min at 4 °C. Following surface staining, cells were subsequently fixed and permeabilized for intracellular Granzyme B detection. Antibody targeting Granzyme B (Cat. 515408, BioLegend, San Diego, CA, USA) was added and left to stain for 45 min at RT. List-mode files from stained samples were acquired a on five-laser Cytek Aurora cytometer for subsequent analysis.

### 3.10. In Vivo AML Studies

P815-mastocytoma/myeloid leukemia cell line was obtained from ATCC. DBA/2j (H-2K^d^) mice were purchased from Jackson Laboratory (Bar Harbor, ME, USA). The mice were maintained by Emory University facilities. Both male and female mice were 8–10 weeks old. On day 0, DBA/2j mice were subcutaneously injected with 1 × 10^5^ P815 cells. The treatments were injected subcutaneously and started from day 7 once daily (ANT308, ANT308C13C17 stp, ANT308C13C17 unstp, and VIP-fully scramble, sc) and once every three days (ANT308-PEG, sc) for 10 days or 14 days. Lymphocyte kinetics were analyzed weekly through pterygoid venous plexus blood collection from the recipients. DBA mice were measured with tumor size by caliper, twice weekly. If either condition was reached or moribund, mice would be euthanized, and the dead would be counted on the next day under the following conditions: when tumor size was over 20 mm on any side or tumor ulceration was >10 mm/infection/necrosis, mouse ability was impaired, unable to eat/drink, or mouse was emaciated. Data were analyzed using Prism version 9 and SPSS statistics 23 are presented as mean ± SD of all evaluable samples if not otherwise specified. Survival differences among groups were calculated with the Kaplan–Meier log-rank test in a pair-wise fashion. Other data were compared using one-way analysis of variance. A *p*-value of <0.05 was considered significant. Animal studies were approved by the Institutional Animal Care and Use Committee (IACUC).

### 3.11. Statistical Analyses

An unpaired *t*-test was performed for comparisons of the two groups. For the comparison of multiple groups, ordinary one-way ANOVA followed by Turkey’s multiple comparison test was used. Tumor size comparison among the groups was carried out with non-parametric analysis. For survival data from mouse studies, the Log-rank test was used. *p* values less than 0.05 were considered significant. All statistical analyses were conducted using GraphPad Prism software, version 10.1.0.

## 4. Conclusions

To enhance the drug properties of ANT308, we employed N-terminus acetylation, peptide stapling, and C-terminus PEGylation. Acetylated ANT308 yielded diminished T cell activation in vitro, suggesting that N-terminus preservation is critical for antagonist activity. Furthermore, modification of residues 13 and 17 proved to be detrimental to antagonist activity, resulting in diminished in vitro activity with no improvement in overall survival and tumor suppression in vivo in a murine AML model. Despite this, the incorporation of an *m*-xylene staple at this position rescued activity and resulted in comparable overall survival and tumor burden to ANT308. The PEGylation of ANT308 proved to be the most effective modification. In vitro studies indicate that a lower concentration of ANT308-PEG better enhanced CD69 and T cell proliferation markers, Granzyme B and Ki67, compared to ANT308. In vivo studies demonstrate that significantly fewer doses of ANT308-PEG were needed to achieve comparable overall survival and tumor burden to ANT308, suggesting that PEGylation increased the biological half-life of the peptide. Plasma stability studies demonstrate that both ANT308 and ANT308-PEG retain the ability to activate human T cells up to 96 h post-incubation in human plasma (Appendix A). However, further pharmacokinetic studies are necessary to evaluate the bioavailability of pegylated ANT308 in contrast to its native form.

## Figures and Tables

**Figure 1 ijms-25-04391-f001:**
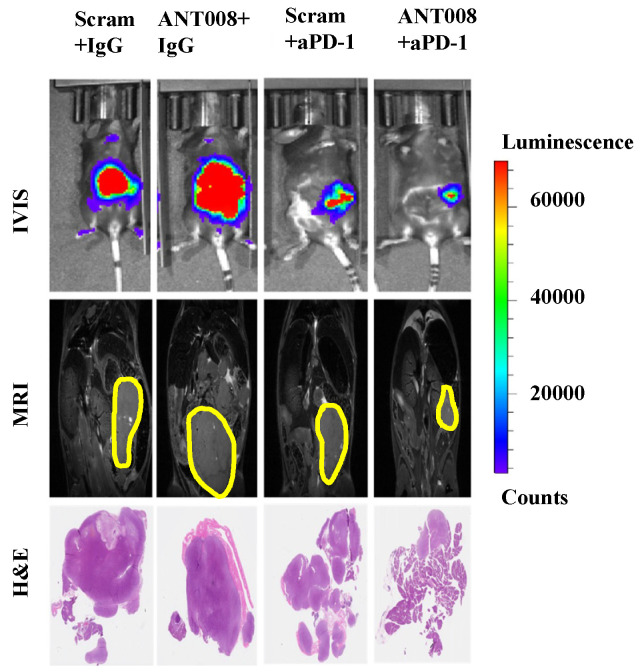
Preliminary tests of an ANT derivative (ANT008) concurrent with immunotherapy (aPD-1) in PDAC murine models demonstrated a significant decrease in tumor growth rate and burden, as well as an increase in CD4+ and CD8+ T cell proliferation. Reprinted with permission from ref. [7,8] under Creative Commons License.

**Figure 2 ijms-25-04391-f002:**
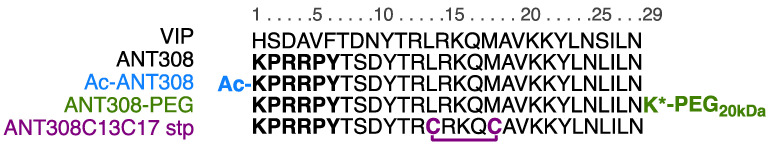
Design of ANT308 derivatives. Residues bolded in black represent key sequence deviation for the antagonist activity of ANT308. Residues bolded in color represent sequence deviation of modified vs. canonical ANT308; * indicates non-canonical azido-lysine.

**Figure 3 ijms-25-04391-f003:**
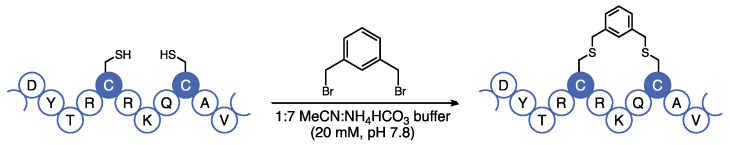
Incorporation of a hydrophobic staple onto ANT308. ANT308C13C17 unstp was synthesized with cysteine at residues 13 and 17; partial sequence shown above. ANT308C13C17 stp was generated post-SPPS via the incorporation of α,α’-dibromo-*m*-xylene under mild conditions.

**Figure 4 ijms-25-04391-f004:**
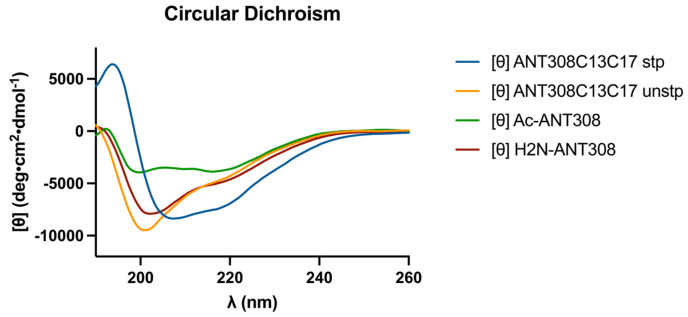
Circular dichroism spectra for *m*-xylene-stapled ANT308, the unstapled control, and acetylated ANT308.

**Figure 5 ijms-25-04391-f005:**
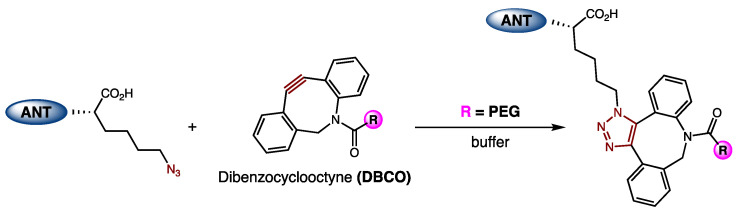
Generation of the ANT308-PEG conjugate via strain-promoted copper-free click chemistry.

**Figure 6 ijms-25-04391-f006:**
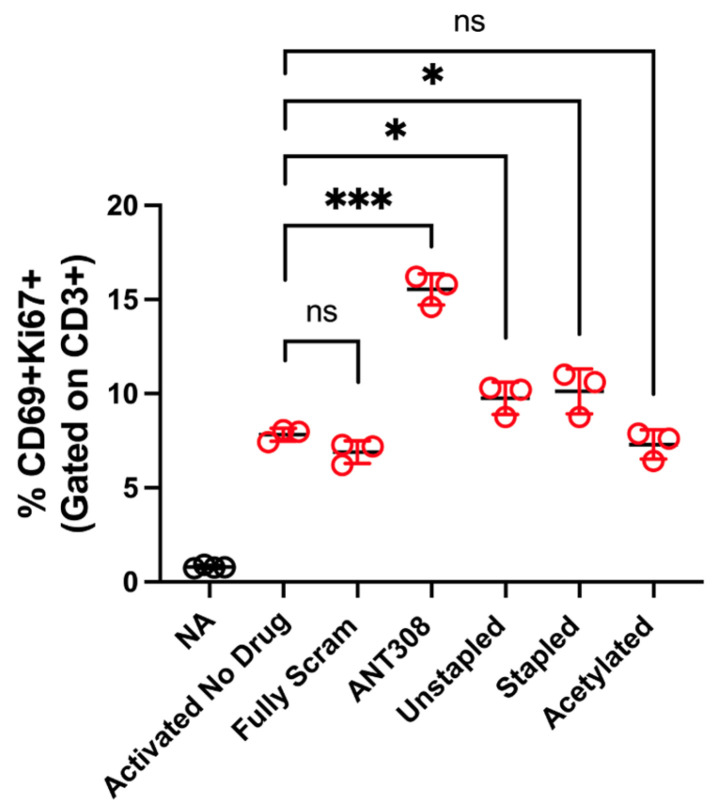
In vitro T cell activation study of acetylated and stapled ANT308. Isolated T cells from four healthy donor PMBCs were pooled and plated in triplicate wells for the assay. T cells were activated with a CD3/CD28/CD2 soluble activator in the absence or presence of peptides for 48 h and assessed for CD69 and Ki67 expression following culture. Activated T cells are in red circles, non-activated T cells are in black circles. The graphs are presented as ± standard deviations. For statistical analysis, an unpaired *t*-test was used. ns is not significantly different, * *p* < 0.05, and *** *p* < 0.001.

**Figure 7 ijms-25-04391-f007:**
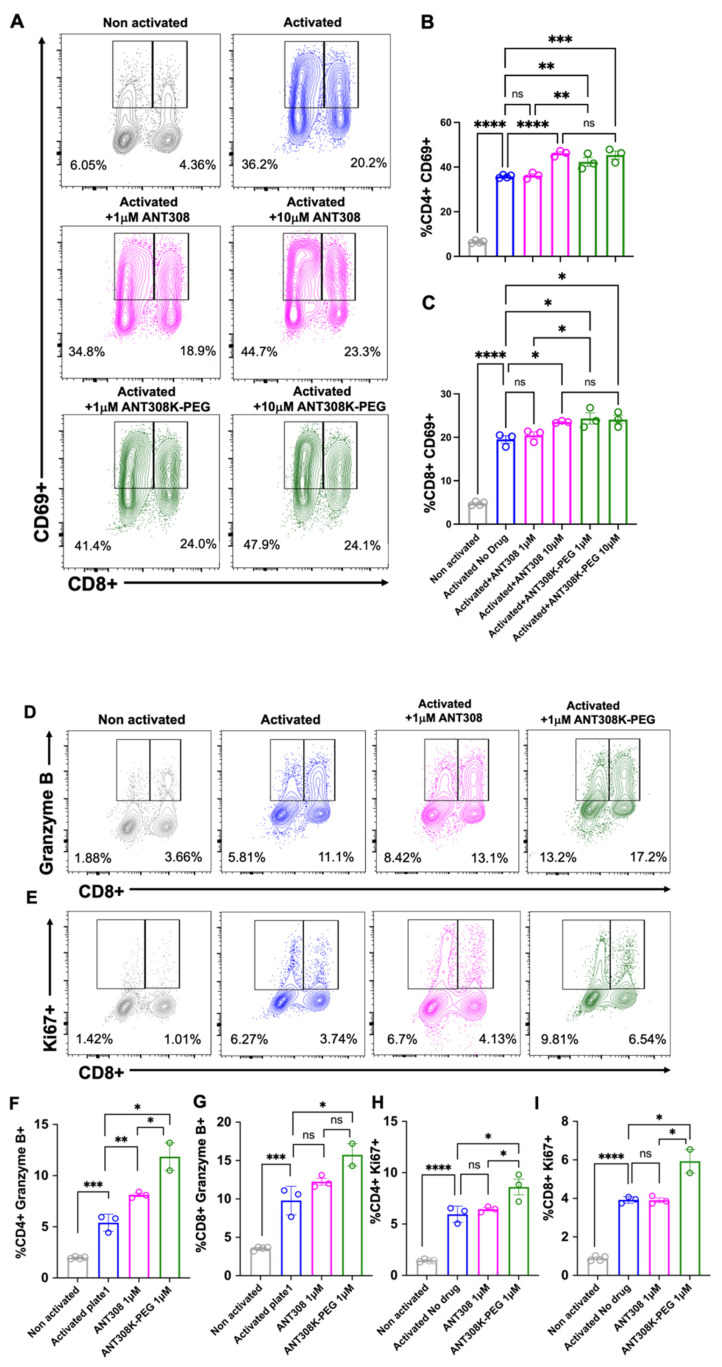
(**A**–**I**): in vitro T cell activation study of ANT308-PEG. (**A**–**C**). ANT308 or ANT308-PEG (1 or 10 μM) were mixed with isolated and pooled human T cells with the presence of a αCD3/CD28 activator and 50 IU/mL IL-2. CD4+ and CD8+ T cell subsets were examined for CD69 expressions 48 h after activation. One-way ANOVA analysis. (**D**–**I**). ANT308 or ANT308-PEG (1 μM) were incubated with pooled human T cells for 48 h; CD4+ and CD8+ T cell subsets were examined for Granzyme B and Ki67 expressions. Representative flow cytometry plots and graphical representations were shown. Two to three replicates each, mean with SEM, unpaired *t* test, * *p* < 0.05, ** *p* < 0.01, and *** *p* < 0.001, **** *p* < 0.0001.

**Figure 8 ijms-25-04391-f008:**
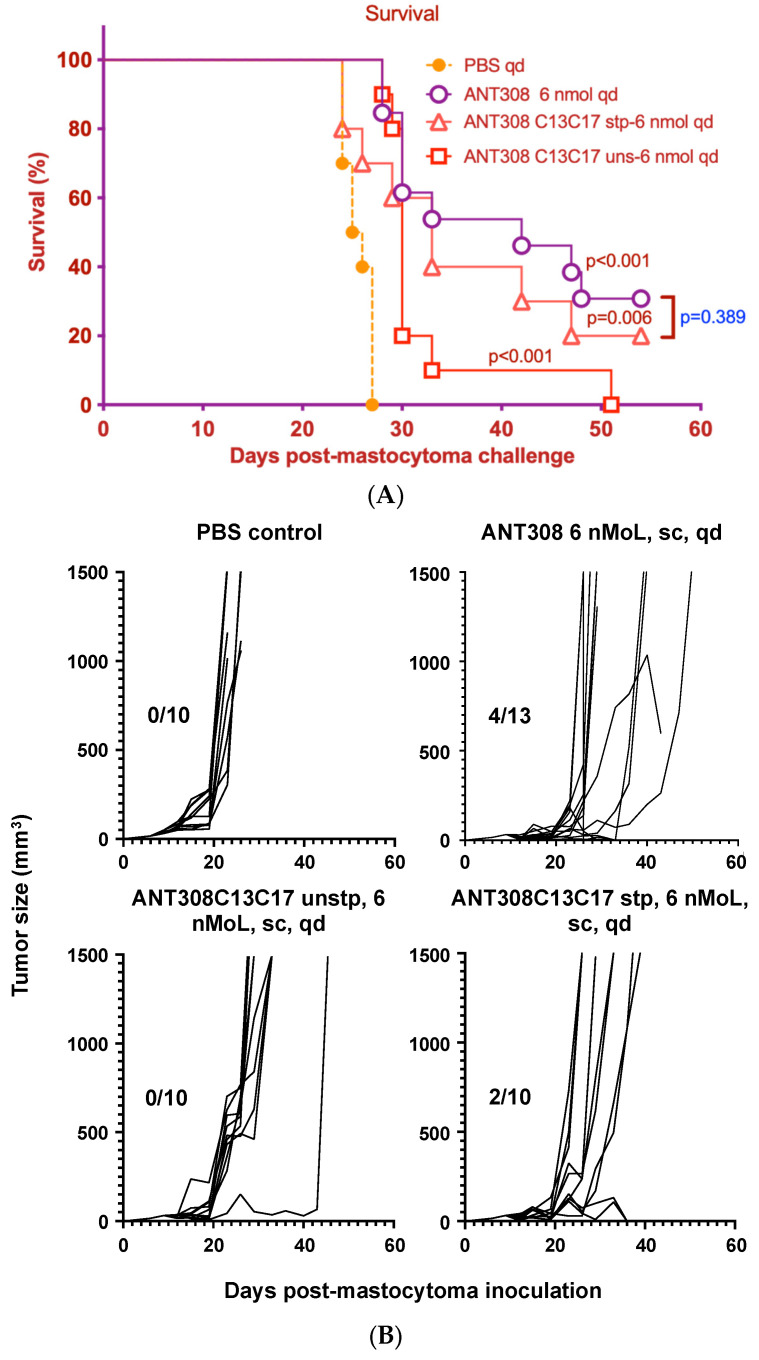
(**A**): In myeloid sarcoma murine models, ten doses of ANT308C13C17 stp had similar prolonged survival of mice with tumor burden to that with ten doses of ANT308, compared to control mice. In 100 μL PBS, 100,000 P815 tumor cells were subcutaneously injected into the right flank after shaving fur. Seven days later, mice were subcutaneously injected with 6 nmol ANT308, 6 nmol ANT308C13C17 unstp, 6 nmol ANT308C13C17 stp, or 200 μL PBS once daily for 10 days. Median survival day (MSD): ANT308—42 days, ANT308C13C17 unstp—30 days, ANT308C13C17 stp—33 days, PBS—26 days. (**B**) In myeloid sarcoma murine models, ten doses of ANT308 control and ANT308C13C17 stp prolonged tumor burden suppression relative to ANT308C13C17 unstp and control mice. In 100 μL PBS, 100,000 P815 tumor cells were subcutaneously injected into the right flank after shaving fur. Seven days later, mice were subcutaneously injected with 6 nmol ANT308, 6 nmol ANT308C13C17 unstp, 6 nmol ANT308C13C17 stp, or 200 μL PBS once daily for 10 days. (The divisor, the number on the left of the division slash, is survival mouse number; the dividend, the number on the right of the division dash, is total mouse number in the group.)

**Figure 9 ijms-25-04391-f009:**
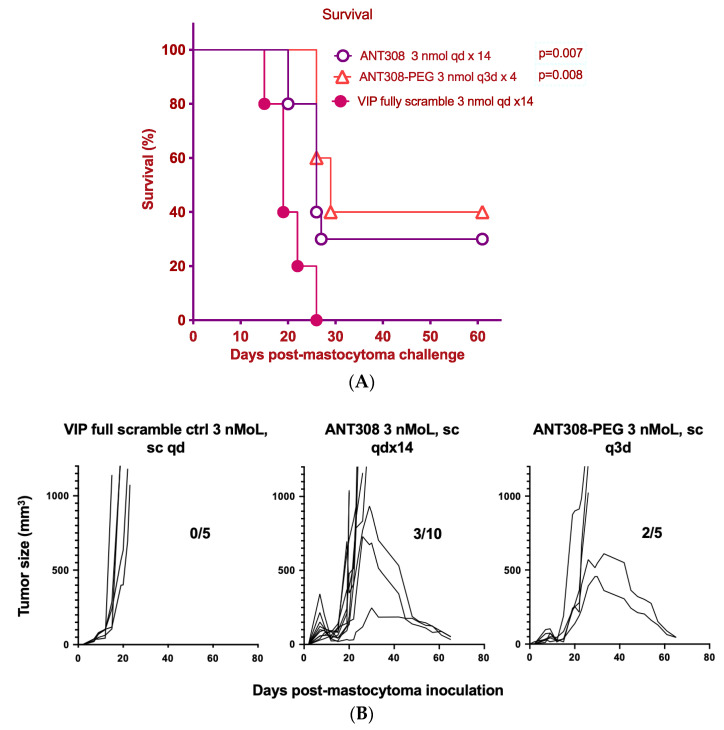
(**A**): Four doses of ANT308-PEG significantly prolonged survival in tumor burden mice, relative to fourteen doses of ANT308, compared with control mice. In 100 μL PBS, 100,000 P815 tumor cells were subcutaneously injected into the right flank after shaving fur. Seven days later, mice were subcutaneously injected with 3 nmol ANT308 once daily, 3 nmol ANT308-PEG once every three days, or VIP full scramble control once daily for 14 days. Median survival day (MSD): ANT308—26 days, ANT308-PEG—29 days, scramble—19 days. (**B**) Four doses of ANT308-PEG significantly prolonged tumor burden suppression relative to fourteen doses of ANT308 and control mice. In 100 μL PBS, 100,000 P815 tumor cells were subcutaneously injected into the right flank after shaving fur. Seven days later, mice were subcutaneously injected with 3 nmol ANT308 once daily, 3 nmol ANT308-PEG once every three days, or VIP full scramble control once daily for 14 days (the divisor, the number on the left of the division slash, is survival mouse number; the dividend, the number on the right of the division dash, is total mouse number in the group).

**Table 1 ijms-25-04391-t001:** Calculated helicity values of the ANT308 derivatives. Calculated using BeStSel [18].

Peptide	α Helicity % ^1^
ANT308	5.30
Ac-ANT308	2.90
ANT308C13C17 unstp	4.84
ANT308C13C17 stp	10.6

^1^ Reference λ 190–250 nm.

**Table 2 ijms-25-04391-t002:** Gradient used to purify ANT308 peptides on RP-HPLC.

Time (min)	Solvent A (%)	Solvent B (%)
0	79	21
5	74	26
19	67	33
24	25	75

**Table 3 ijms-25-04391-t003:** List of antibodies used for flow cytometry.

Target	Fluorochrome	Vendor	Catalog No.
CD3	PE/Cyanine 5	BioLegend	300410
CD4	APC-Cy7	BD	557871
CD8	Alexa Fluor 700	BD	557945
CD69	Brilliant Violet 650	BioLegend	310934
4-1BB	Brilliant Violet 650	BioLegend	309828

## Data Availability

All data generated and analyzed in this study are included in this publication and its corresponding Appendix A.

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
