# Peer review of "Chemical Modifications to Enhance the Drug Properties of a VIP Receptor Antagonist (ANT) Peptide"

_ijms, 2024, doi:10.3390/ijms25084391_

Round 1

Reviewer 1 Report

Comments and Suggestions for Authors

GENERAL COMMENTS 

1. The authors could smoothen the CD spectra for better clarity and they could mention the concentration of each system under study, also mention whether the variaton in the buffer and pH of the medium of each system will exhibit any variation in the MRE 

2. Figure5: Scheme mentions the most mechnanistic pathway, did the authors observe any side products?

Reviewer 2 Report

Comments and Suggestions for Authors

In this work the Authors present the results of their efforts to enhance the drug properties of ANT308 by chemical (covalent) modifications such as N-terminus acetylation, peptide stapling, and PEGylation. The submitted manuscript is of high quality and should be published after minor revisions. The work is clear, concise, present the significant amount of newly obtained results.

Continuous line numbering would really facilitate the review process. Please don’t forger about this next time.

Page 2, “Because peptide therapeutics can possess desirable physiochemical properties of both small molecule and biologics” – what exactly properties do you have in mind? Honestly, I can’t find any.

While the Authors have performed the statistical analysis, they haven’t described it in the materials and methods section. This should be corrected by adding the information about the software and tests used.

The Authors have created Figures 3 and 5 to show the reaction of two out of three chemical modification of an API. While acetylation is more simple than the other two, its scheme should be also introduced in a form of a figure.

While this was not the aim of the study, the Authors could have used the molecular modeling methods to assess how the modification of the peptide would affect the binding affinity. Nowadays, peptide-to-protein molecular docking methods are quite advanced and can be used to predict the outcomes of the experiment, which would allow to more efficient planning. I.e., look here: 10.3390/molecules29010272 .

Also, the molecular dynamics methods can be used to correlate the structure with the CD-spectra for more accurate estimation of peptide secondary structure.

Besides, in the future studies, I highly recommend to use the NMR in solution to better characterize the peptide structure.

Reviewer 3 Report

Comments and Suggestions for Authors

The manuscript ijms-2952253 reports the design, synthesis, and biological evaluation of a peptide. The manuscript is well written, and I recommend the publication after minor revisions as follows below:

1) In the circular dichroism analysis provide the normalized spectra and use the online server to determine the structural content percentages. 

2) Since the authors measured the CD signal below 200 nm it is not clear if they used or not an inert atmosphere. Please, clarify it in the methodology section.

3) Please, provide the high resolution mass spectrometry data as Supplementary Material. This is important to full characterize the peptide.

4) What is the reason that the authors only evaluated the FAR-UV CD and not also the NEAR-UV-CD? I recommend that the authors also evaluate the NEAR-UV-CD to better understand the three-dimensional structure of the peptide.

5) Since the synthetic peptide was also designed to improve the stability in the chemical matrix, please, conduct chemical-physical and biological stability assays for the synthetic peptide.
